# Precise prediction of phase-separation key residues by machine learning

Jun Sun[1,2,3,4,5,9], Jiale Qu[3,4,5,9], Cai Zhao[3,4,5,9], Xinyao Zhang[3,4,5], Xinyu Liu[3,4,5], Jia Wang[3,4,5,6], Chao Wei[3,4,5], Xinyi Liu[3,4,5], Mulan Wang[3,4,5], Pengguihang Zeng[3,4,5], Xiuxiao Tang[3,4,5], Xiaoru Ling[3,4,5], Li Qing[3,4,5], Shaoshuai Jiang[3,4,5], Jiahao Chen[3,4,5], Tara S. R. Chen[7], Yalan Kuang[1,2], Jinhang Gao[1,2], Xiaoxi Zeng[1,2], Dongfeng Huang[7], Yong Yuan[1,2] ✉, Lili Fan [8] ✉, Haopeng Yu[1,2] ✉ & Junjun Ding [1,2,3,4,5,7] ✉

Understanding intracellular phase separation is crucial for deciphering transcriptional control, cell fate transitions, and disease mechanisms. However, the key residues, which impact phase separation the most for protein phase separation function have remained elusive. We develop PSPHunter, which can precisely predict these key residues based on machine learning scheme. In vivo and in vitro validations demonstrate that truncating just 6 key residues in GATA3 disrupts phase separation, enhancing tumor cell migration and inhibiting growth. Glycine and its motifs are enriched in spacer and key residues, as revealed by our comprehensive analysis. PSPHunter identifies nearly 80% of disease-associated phase-separating proteins, with frequent mutated pathological residues like glycine and proline often residing in these key residues. PSPHunter thus emerges as a crucial tool to uncover key residues, facilitating insights into phase separation mechanisms governing transcriptional control, cell fate transitions, and disease development.

Liquid-liquid phase separation (LLPS) is one of the most important biophysical mechanisms mediating the formation of membraneless compartments from macromolecules, such as proteins and nucleic acids[1]. Over the past decade, LLPS of biomolecules has been established as a unifying physical mechanism underlying transcriptional control[2–4], autophagy[5], chromatin formation[6,7] and chromatin structure organization[8–10]. While certain sequence features of proteins, such as modular interacting domain and low complexity sequences, that promote LLPS have been well studied, how small subset of amino acids are quantitatively encoded to contribute to phase separation remains largely unknown[11–13], which limits the functional study of phase separation.

At present, the investigation of phase separation functionalities involves altering core regions or residues. Considering the strong correlation between the intrinsically disordered regions (IDRs) of phase-separating proteins and their phase separation capacity[14,15], truncating IDRs can disrupt protein phase separation[16–19]. Additionally, substitutions within charged or multivalent interaction centers,

[1]Department of Thoracic Surgery and West China Biomedical Big Data Center, West China Hospital, Sichuan University, Chengdu 610041, China. [2]Med-X Center for Informatics, Sichuan University, Chengdu 610041, China. [3]RNA Biomedical Institute, Sun Yat-sen Memorial Hospital, Zhongshan School of Medicine, Sun Yat-sen University, Guangzhou, Guangdong, China. [4]Advanced Medical Technology Center, The First Affiliated Hospital, Zhongshan School of Medicine, Sun Yat-sen University, Guangzhou, Guangdong, China. [5]Center for Stem Cell Biology and Tissue Engineering, Key Laboratory for Stem Cells and Tissue Engineering, Ministry of Education, Zhongshan School of Medicine, Sun Yat-sen University, Guangzhou, Guangdong, China. [6]GMU-GIBH Joint School of Life Sciences, Guangzhou Medical University, Guangzhou 511436, China. [7]Department of Rehabilitation Medicine, The Seventh Affiliated Hospital, Sun Yat-Sen University, Shenzhen, Guangdong 518107, China. [8]Guangzhou Key Laboratory of Formula-Pattern of Traditional Chinese Medicine, School of Traditional Chinese Medicine, Jinan University, Guangzhou, Guangdong, China. [9]These authors contributed equally: Jun Sun, Jiale Qu, Cai Zhao. ✉e-mail: yongyuan@scu.edu.cn; fanlili@jnu.edu.cn; yuhaopeng@wchscu.cn; dingjunj@mail.sysu.edu.cn

prevalent in IDRs[20–27] (such as replacing them with amino acids like alanine), can influence protein phase separation dynamics[28,29]. However, these alterations often encompass sizable protein fragments or multiple residues, inevitably impacting not only their phase separation properties but also their non-phase separation functions. Thus, the identification of 'key residues' becomes imperative – residues that, with minimal alteration, exert maximal influence on phase separation dynamics, enabling precise modulation while minimizing disruption to non-phase separation functionalities.

Several diseases, including neurodegenerative disorders, skin diseases, syndactyly syndrome and cancers, are associated with phase separation[30–35], although the exact mechanisms are unknown. Since pathological mutations can perturb phase separation, the latter can be a useful biomarker of diseases with complex genetic basis[36]. However, without quantifying the contribution of specific amino acids within the small subset (key residues) to phase separation, it is challenging to screen the core mutations affecting phase separation from numerous clinical mutations. Therefore, it is necessary to identify pathological mutations within key residues that can impact phase separation.

To this end, we developed Phase-Separating Protein Hunter (PSPHunter), a machine-learning algorithm that can predict phase-separating proteins and identify key residues by integrating the protein sequences and functional features from existing resources of phase-separating proteins. This tool was further used to quantify the impact of disease-related mutations on phase separation and provide an analytical framework to dissect the relationship between phase separation and diseases.

## Results

### Establishment of PSPHunter for predicting phase-separating proteins

To identify essential key residues for studying phase separation functions and related disease mechanisms, we initially established PSPHunter, a machine learning algorithm for predicting phase-separating proteins (Fig. 1a). We systematically constructed eight distinct datasets, each designed to meet specific criteria, detailed extensively in Supplementary Fig. 1a, b, Supplementary Table 1, and the Methods section. These datasets encompassed training sets integrating mixed-species data namely MixPS237 and MixPS488. Additionally, we compiled human-specific phase-separating proteins from four public databases (PhaSepDB[37], LLPSDB[38], DrLLPS[39], PhaSePro[40]) and relevant literature sources (Supplementary Fig. 1a). This curation effort culminated in the compilation of a set comprising 167 human phase-separating proteins (hPS167).

In order to extract more comprehensive features, we collected both sequence and functional features that might be attributed to phase separation (Supplementary Fig. 1b, c). Given the prominence of the sticker-spacer model as a representative phase separation mechanism[41], we incorporated word2vec feature during the design of sequence attributes. The word2vec feature quantifies the combinations of short sequence segments within protein sequences, analogous to how words encode meaning in human language. We believe that word2vec features can encode the grammar of phase separation due to their exceptional performance in predicting protein phase-separating capacity (Fig. 1b). We also introduced Position-Specific Scoring Matrix (PSSM) and Hidden Markov Model (HMM) features to capture protein evolutionary traits, projected secondary structure, and accessible residues with relative solvent accessibility for structural insights. Alongside, our methodology encompassed functional traits, spanning modifications, mutations, network properties, protein abundance, and other pertinent characteristics (Supplementary Fig. 1c).

Regarding the sequence features, the phase-separating proteins are larger, and have more post-translational modification sites, along with larger IDR, RNA and DNA binding regions compared to the non–phase-separating proteins (Supplementary Fig. 1d). In addition,

the phase-separating proteins also have a higher proportion of glycine and proline residues as opposed to hydrophobic residues (Supplementary Fig. 1d). In terms of functional features, phase-separating proteins are more abundant, have more evolutionarily conserved features, and have important network characteristics (Supplementary Fig. 1e). Notably, individual functional features demonstrate predictive capabilities (Supplementary Fig. 1g). This aligns with our feature importance analysis, which highlights the pivotal roles of network properties such as protein closeness in the protein-protein interaction network and phosphorylation levels in protein post-translational modifications in the classification process (Supplementary Fig. 1i). Further analysis of feature correlations reveals a marked internal correlation between sequence and functional features compared to their intercorrelations (Supplementary Fig. 1h).

By integrating all 123 sequence-based and functional attributes, their mutual complementarity significantly enhances the classification performance in predicting phase-separating proteins (Fig. 1b, Supplementary Table 2, Supplementary Table 3). Moreover, based on the insights gained from feature importance analysis, we found that the top 60 ranked features yielded comparable results to using the complete feature set (Fig. 1c). Subsequently, we employed this feature-selected model to predict all human-reviewed proteins, and remarkably, the resulting predictions exhibited strong agreement with those generated using the full feature set (Supplementary Fig. 1j, k), underscoring the robustness of our methodology.

Additionally, we conducted a comprehensive systematic assessment involving varying data sizes with different levels of redundancy (sequence similarity below 30%, 60%, 90%), diverse feature integration strategies (two-layer stacking, averaging probabilities from different sub-classifiers, direct combination of all features), and multiple machine learning methods (support vector machine, naïve Bayesian classifier, neural network, random forest, light gradient boosting machine, and extreme gradient boosting) for model construction (Fig. 1d and Supplementary Fig. 2a–c). Ultimately, we adopted an approach that maintains sequence similarity among samples in the training set below 30%, directly integrates sequence and functional attributes, selects the top 60 importance-ranked features, and utilizes the random forest algorithm to characterize our final predictive performance.

In comparison with other prediction models on Independent_Test_III_hPS167, PSPHunter demonstrates advantages in the prediction of phase-separating proteins by integrating multifaceted information (Fig. 1e, Supplementary Fig. 2d and Supplementary Table 4). The robustness of the PSPHunter algorithm is also evident in the mixed-species training datasets MixPS237 and MixPS488 (Supplementary Table 5). Despite the dataset expansion and the incorporation of mixed-species information, PSPHunter's predictive capabilities show minimal impact. When extending predictions to the entire proteome, 70% of phase-separating proteins are successfully replicated (Supplementary Fig. 2e, f). Additionally, the model's predictive scores exhibit a notably high correlation (hPS167 vs MixPS488: PCC = 0.774; hPS167 vs MixPS237: PCC = 0.701; Supplementary Fig. 2g). These findings collectively underline PSPHunter's strength in predicting phase-separating proteins and its robustness across diverse datasets.

### PSPHunter can predict reliable phase-separating proteome

To assess the phase separation capacity of all human proteins, we introduced the PSPHunter score, which is derived from PSPHunter's phase-separating proteins prediction model. To validate the credibility of the PSPHunter score, we computed its distribution within four-tier protein datasets and observed that the ranking correlated consistently with weighted experimental evidence[42] (Fig. 1f). Compared to random proteins, typical phase-separating proteins such as processing bodies, stress granules, disordered proteins and RNA-binding proteins had a

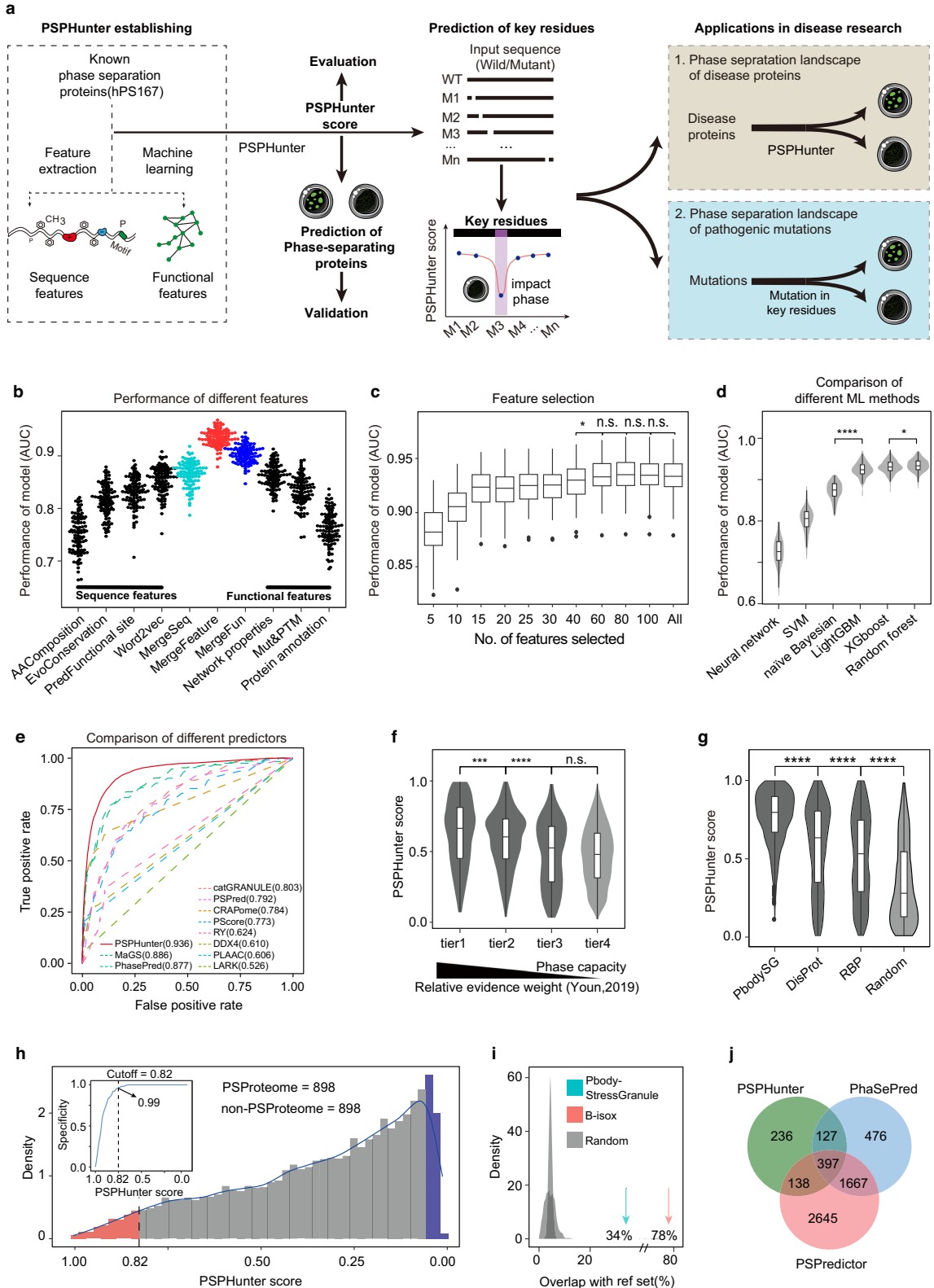

significantly higher capacity of phase separation as per the algorithm (Fig. 1g). All these results indicated that PSPHunter score can be used for identify potential phase-separating proteins.

To gain a more reliable phase-separating proteome (PSProteome), we optimized the threshold of PSPHunter score by independent testing (Fig. 1h). The results suggested that a PSPHunter score cutoff of 0.82 was proper, as its corresponding specificity was

0.99. Then, a total of 898 phase-separating proteins were screen out, constituting the PSProteome, out of which 747 were newly predicted (Supplementary Data File 1). In addition, 78% of the RNA granules identified by the b-isox method[43] was covered by the PSProteome (Figs. 1i) and 73.5% of the PSProteome was reproduced by PhaSePred[44] and PSPredictor[45] (Fig. 1j and Supplementary Fig. 2h). Functional analysis showed that the PSProteome is enriched in phase

**Fig. 1 | Establishment of PSPHunter for predicting phase-separating proteins.**
**a** Scheme of the research route and methods. PSPHunter incorporates sequence and functional features for phase-separating protein prediction and key residue identification, enabling exploration of the link between phase separation and diseases. **b** Feature performance comparison: Sequence and functional features (cyan and blue, respectively) are compared. Combining these features significantly improves prediction accuracy ($n = 100$ datasets). **c** Model performance evaluation based on feature importance ($n = 100$ datasets), considering varying numbers of selected features. **d** Violin plots comparing the performance among different machine learning methods ($n = 100$ datasets), including support vector machine (SVM), naïve Bayesian classifier (NB), neural network (NN), random forest (RF), light gradient boosting machine (LightGBM), and extreme gradient boosting (XGBoost). **e** Comparison of PSPHunter with other representative phase-separating protein predictors. We randomly extracted 30% of data from the positive samples, and an equally sized set of negative samples was selected to form the independent test dataset. Employing this selection strategy, we created 100 distinct independent test sets. The final evaluation represents the average performance across all sets. **f** Violin plots illustrating PSPHunter scores of four-tier protein datasets ranked based on weighted experimental evidence according to Youn et al., tier1 = 367, tier2 = 473, tier3 = 426, tier4 = 3111. **g** PSPHunter score distribution in different phase separation-related datasets, processing bodies and stress granules (from Wikipedia, $n = 591$), disordered proteins (from DisProt, $n = 568$), and RNA binding proteins (from EuRBPDB, $n = 1784$). **h** PSPHunter scores in the proteome: Candidate phase-separating proteins (PSProteome, red) and non-phase-separating proteins (blue) are identified based on a sequence identity cutoff ($n = 898$ each). **i** Overlap with reference datasets: Overlap between the PSProteome and datasets identified by the b-isox method (red) and processing bodies/stress granules (cyan) is shown. **j** Overlap between the PSProteome and the latest two-phase separation predictors. Note: All statistical tests used one-sided Wilcoxon tests. Significance levels are: *$P < 0.05$, **$P < 0.01$, ***$P < 0.001$, ****$P < 0.0001$ (n.s. = not significant). Boxplots represent the interquartile range (IQR) from Q1 to Q3, with the median as the middle line. Whiskers extend up to 1.5 times the IQR. Outliers are not shown.

separation-related processes such as RNA processing and regulation[29,46] (Supplementary Fig. 2i).

To further benchmark PSPHunter's ability to predict phase-separating proteins, we established three independent test sets, including mixed-species, human and non-human datasets. Benchmarking results demonstrate that our model exhibits better performance in the human dataset and yields good performance in previously unseen datasets from other species (Fig. 2a and Supplementary Table 6). Moreover, we performed in vitro and in vivo experiments to test our prediction results, by using both fluorescence labeling and fluorescence recovery after photobleaching (FRAP). We selected a benchmark dataset comprising 12 proteins with the highest PSPHunter scores (potential new phase-separating candidates within PSProteome), along with 5 proteins with the lowest scores (non-phase-separating protein contrasts) as contrast (Supplementary Data File 1). The candidate phase-separating proteins formed green puncta in vitro, and FRAP was consistent with their dynamic nature (Fig. 2b, c and Supplementary Fig. 3a, d). In contrast, the non-phase-separating formed irregular shapes and the fluorescence was difficult to recover after photobleaching. Similar phenomena were observed in the in vivo experiments as well (Fig. 2b, d and Supplementary Fig. 3b, d). In addition, we also observed phase separation fusion events both in vitro and in vivo for the newly identified phase-separating proteins (Supplementary Fig. 3c). Thus, all the selected 12 predicted phase-separating proteins behave as liquid droplets. Again, quantification experiments (Fig. 2e) and correlation analysis (Fig. 2f) demonstrate that PSPHunter accurately measures the phase separation capacity of proteins.

Taken together, the above results demonstrate that the PSPHunter score can effectively characterize the phase separation capacity of proteins. PSPHunter, in turn, provides a reliable landscape of phase-separating proteome.

## PSPHunter can accurately predict key residues of phase-separating proteins

Precision manipulation of phase separation is essential to understand its role in transcriptional control[2–4], cell fate transitions[9,10], and disease development[15,32–34,47–50]. Therefore, it is crucial to identify the specific subset of amino acids exerting the most pronounced influence on phase separation, a classification termed herein as 'key residues'. Disrupting these residues can impact fundamental properties of protein phase separation, including the condensates' liquid-like behavior and saturation concentration. Amino acid sequences embed the phase separation-related features[13,26,51], suggesting that perturbing the sequence composition could reveal these key residues.

To systematically screen key residues, we design features to enhance PSPHunter's sensitivity to variations in amino acid sequences. Then we aimed to study the phase separation influence of each amino

acid by introducing a sliding-window strategy (Fig. 3a, Supplementary Fig. 4a). The full-length proteins were sequentially truncated, and PSPHunter was employed to compute the phase-separation capacity for each residue-deleted protein. Lower scores indicated a greater influence of the residue on phase separation. We identified the residues that exhibited the most substantial change in phase separation capacity upon truncation as the key residues (see the 'Methods' section for detailed information). Finally, the contribution of all residues to phase separation was globally scanned, and the consecutive key residues that formed valleys were considered as key regions (Fig. 3a).

To validate the reliability of PSPHunter in predicting key residues of phase-separating proteins, we compared the key regions identified by PSPHunter with known phase-separating regions sourced and benchmarked from the PhaSePro database (https://phasepro.elte.hu)[40] (Fig. 3b, Supplementary Data File 2). PSPHunter reproduced 105 out of 144 known phase-separating regions (Fig. 3b). In comparison to random fragments, the key regions identified by PSPHunter were notably closer to the known phase-separation regions (Fig. 3c). Yang et al. reported that the intrinsically disordered regions (NTF2) and RNA-binding domain (RRM) of G3BP1 contribute to its phase separation capacity[52]. The PSPHunter prediction not only reproduced these specific regions but also narrowed them into a few residues, which may provide greater insights into how specific amino acids affect phase separation. Furthermore, we were able to demonstrate that coiled-coil (CC) region of YAP[53], glycine-rich region of TDP43[17], the linker connecting the first two SH3 domains of NCK1[54], SH3 domain of GRB2[55], and low-complexity domain of FMRP[56] were the key regions of the respective proteins, which was consistent with previous studies (Fig. 3d and Supplementary Fig. 4b). Identification of the fine key regions of other typical phase-separating proteins can improve our understanding of their role in phase separation (Supplementary Fig. 4b). Collectively, it provides a continuous score for each amino acid, PSPHunter can identify the key residues for phase separation.

## Key residues predict by PSPHunter can be used for precise manipulation of phase separation

To further experimentally confirm whether the predicted key residues can indeed manipulate phase separation, we also designed non-key residues based on the hypothesis that truncation of the key residues rather than the non-key residues would impair phase separation (Fig. 3e and Supplementary Fig. 4d, e). FRAP analysis demonstrated that truncation of the non-key residues of GATA3 (88–93 aa, Mut-Control) had little effect on phase separation, while that of only six key residues of GATA3 (322–327 aa, Mut-Key) significantly reduced the dynamic behavior of the corresponding puncta both in vitro and in vivo, along with the decrease in the number of GATA3 puncta (Fig. 3f–i). Additionally, we conducted an assessment of the saturation

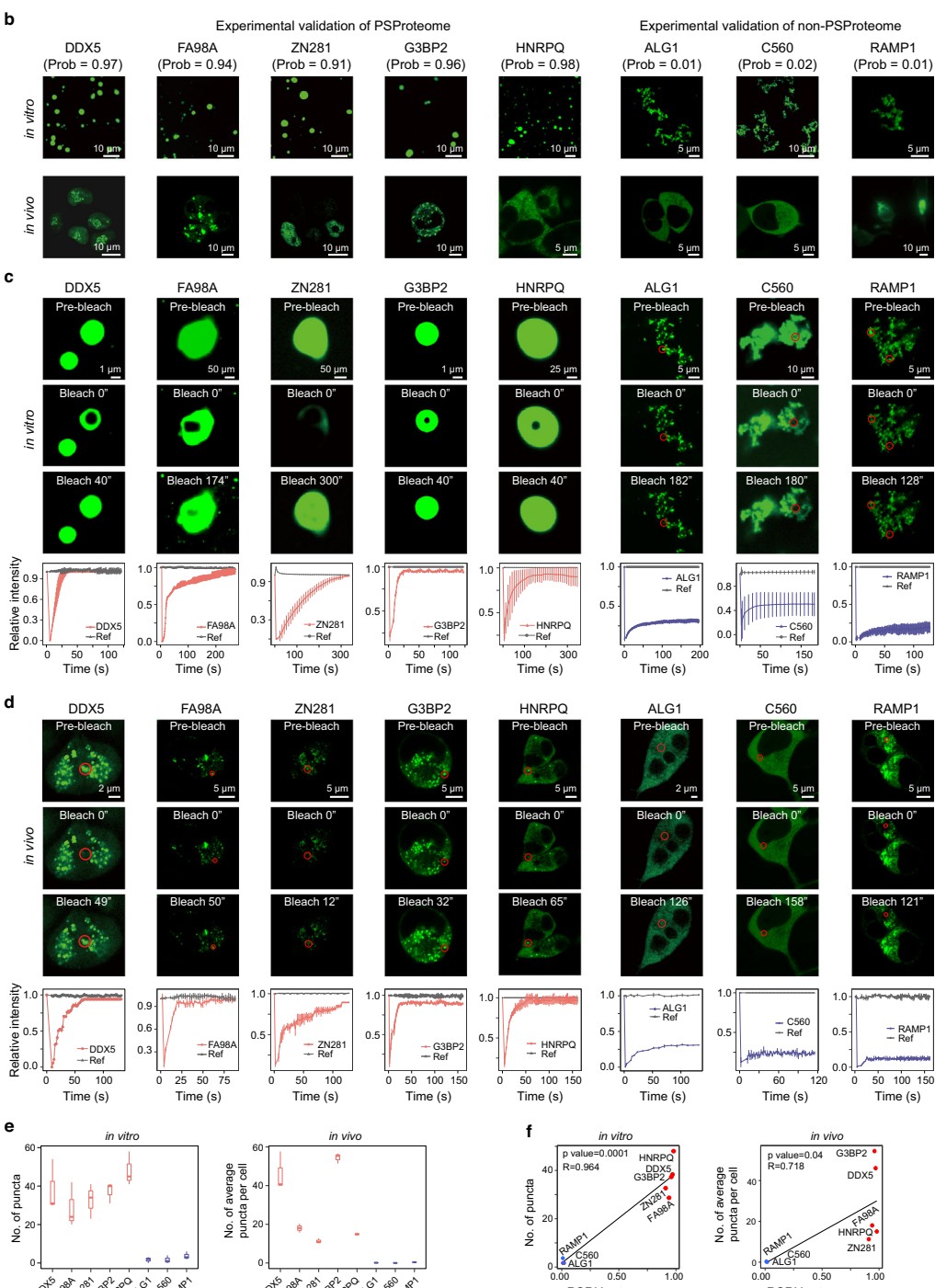

| Model | Dataset | Recall | Precision | F1 score | ACC | MCC | AUC |
|---|---|---|---|---|---|---|---|
| Training_I_MixPS488 | Independent_Test_I_MixPS488 | 0.849 | 0.819 | 0.833 | 0.830 | 0.662 | 0.911 |
| Training_II_MixPS237 | Independent_Test_II_MixPS237 | 0.851 | 0.836 | 0.842 | 0.840 | 0.683 | 0.905 |
| Training_III_hPS167 | Independent_Test_III_hPS167 | 0.854 | 0.868 | 0.857 | 0.861 | 0.725 | 0.933 |

concentration for different GATA3 mutants. The truncation of key residues within GATA3 necessitates relatively higher concentrations to form phase separation. Under these elevated concentration conditions, the observed puncta formation is comparatively diminished (Supplementary Fig. 4f, g). These GATA3 outcomes collectively illustrate that key residues play an important role in the capacity of protein phase separation. Furthermore, the PSPHunter algorithm was applied

successfully to predict the key residues of the core pluripotency factor OCT4[9] (truncated 3 residues), SOX2 (truncated 6 residues, Supplementary Fig. 4c, d) and the PcG family protein RYBP[10] (truncated 21 residues) in our recent studies.

In summary, these findings collectively illustrate that PSPHunter is capable of identifying residues for precise manipulation of phase separation.

**Fig. 2 | PSPHunter can predict reliable phase-separating proteome. a** Evaluation of PSPHunter model performance across different independent test sets. Among these, 'Independent_Test_I' and 'Independent_Test_II' represent test sets comprising a mixture of species sourced from 'MixPS488' and 'MixPS237', respectively. Additionally, 'Independent_Test_III' encompasses a human-specific dataset obtained from 'hPS167'. For further details, refer to the methodology section. **b** in vitro and in vivo validation of potential phase-separating proteins. Red denotes top-ranked proteins with high PSPHunter scores, indicating their potential as phase-separating proteins, while blue denotes bottom-ranked proteins with low PSPHunter scores, serving as negative controls. **c, d** in vitro (**c**) and in vivo (**d**) FRAP analysis of potential phase- separating proteins. For each protein, we quenched

three spots for statistical analysis of droplet properties (The error bars represent the standard deviation). The droplets formed by potential phase-separating proteins exhibit rapid recovery from photobleaching, while the negative controls display limited recovery. **e** Quantification of puncta for potential phase-separating and non-phase-separating proteins in vitro and in vivo, $n = 3$, the boxplots were drawn from lower quartile (Q1) to upper quartile (Q3), with the middle line denoting the median, whiskers with maximum 1.5 interquartile range (IQR) and outliers were not indicated. **f** Correlation between PSPHunter scores and the number of puncta in vitro and in vivo experiments, Pearson's product-moment correlation coefficient, two-sided.

## Glycine and its motifs are enriched in spacer and key residues

To discover the characteristics of key residues, we constructed the phase-separating key residues landscape in human PSProteome (Fig. 4a). We found that the majority of phase-separating proteins contain 3–4 key regions ranging from 21 to 115 amino acids (Fig. 4b and Supplementary Fig. 5a), and are more likely to localize in IDRs (Fig. 4c) rather than nucleic acid-binding regions (usually folded domains). However, nearly one-third of key residues were predicted to be in the nucleic acid-binding domains, which possess limited independent phase separation capacity (Fig. 4c, d). This observation highlights the fact that both disordered and folded domains contribute to phase separation[57].

Previous studies have highlighted the relevance of specific amino acids to phase separation, including the enrichment of glycine (G) and proline (P) in phase-separating proteins[58–61], along with the crucial role played by aromatic amino acids in the typical spacer-sticker model[41,62]. To systematically explore the impact of amino acid enrichment on the formation of liquid-phase structures, we evaluated the prevalence of 20 amino acids within key residues. Our findings confirmed the enrichment of GP (Fig. 4e, f). Both G and P amino acids predominantly serve as spacers[63], and they are relatively more abundant in phase-separating proteins (Fig. 4h and Supplementary Fig. 5c). These spacers are typically found in the intrinsically disordered regions (IDRs) rather than in the folded regions of phase-separating proteins to facilitate droplet mobility[14,64], which align with our observation (Fig. 4h).

Earlier research has also elucidated the influence of specific amino acid motifs within the spacer-sticker model[12]. In our investigation, we observed the presence of G motifs in both phase-separating proteins and key residues (Fig. 4g and Supplementary Fig. 5b), likely due to the proximity of spacers (Fig. 4j and Supplementary Fig. 5h). However, we did not detect polyP motifs in key residues. This difference may be attributed to the unique properties of proline in protein structure and other relevant factors. Proline exhibits unique structural features[58,65] within phase-separating proteins, hindering the formation of conventional secondary structures. Over 80% of proline residues are located in the coil regions of secondary structures (Supplementary Fig. 5d). In our previous study[9], the experimental results suggest that two segments of GP-enriched key residues predicted by PSPHunter in the OCT4 protein are located in the turn or bend regions of IDRs (Supplementary Fig. 5f, g). Truncating these proline residues significantly impacts the liquid-like behavior OCT4.The disappearance of spacer such as proline might lead to a loss of relatively stable structures and causing disruptions in the multivalent interactions, including aromatic systems[66,67].

Regarding the broader concept "stickers" (aromatic FWY and RLDM residues[41]), despite their well-documented significance in protein phase separation, these amino acids are relatively less abundant in proteins undergoing phase separation, with their distribution primarily concentrated in the folded regions of proteins (Fig. 4i and Supplementary Fig. 5e). Notably, the sticker residues, may not need to be continuously present (Supplementary Fig. 5h). Even scattered occurrences in sequences can contribute to phase separation.

Taken together, we systematically explored key residues within phase-separating proteins, revealing the significance of glycine and proline as spacers and their role in promoting protein phase separation (Fig. 4k) and PSPHunter's approach uncovered a wider spectrum of phase-separation mechanisms among key residues.

## The pathogenic mutations glycine and proline disrupt phase separation more significantly than other mutations

Studying the relationship between diseases and phase separation offers insights into the regulation of phase separation and the mechanisms of disease[15,32–34,47–50]. To assess the association between PSPHunter score and diseases, we computed the distribution of PSPHunter scores in previously reported diseases related to phase separation[68]. The results revealed that, compared to randomly selected diseases, Mendelian diseases and cancers exhibit significantly higher PSPHunter scores (Fig. 5a). Disease-related proteins exhibit a notable enrichment among phase-separating proteins (Fig. 5b). Remarkably, these proteins constitute nearly 80% of the PSProteome (Fig. 5c). Given that a single gene can be associated with multiple diseases, our analyses unveil a positive correlation between a protein's propensity for phase separation and the number of diseases it is linked to or the quantity of associated mutations it harbors (Supplementary Fig. 6a–d). In particular, we collected mutations known to either promote or inhibit phase separation, such as the introduction of alanine substitutions in HOXD13[32] (associated with hereditary synpolydactyly), the mutation of charged residues to alanine in OCT4[9], the phenylalanine-to-alanine mutation in DDX4[28], and the fusion IDR to mutated proteins[9]. The predicted results of PSPHunter show good agreement with previous reports (Supplementary Fig. 6e–g). These findings collectively suggest a substantial interconnection between phase separation caused by pathogenic mutations and disease etiology.

We further applied PSPHunter to evaluate the phase separation effects of genetic mutations. 36,413 missense mutations including 33,729 pathological mutations and 2684 neutral variants associated with PSProteome were collected from a comprehensive database (HuVarBase[69]). It shows that phase-separating proteins have more missense mutations than non-PS proteins (Fig. 5d). In addition, the phase-separating proteins harbored more pathological mutations rather than neutral mutations (Fig. 5e and Supplementary Fig. 6h), suggesting that the former may be responsible for the dysregulated phase separation. Using PSPHunter, we found that pathological mutations had a greater effect on protein phase separation capacity than neutral mutations (Fig. 5f), especially the point mutations of VHL, ESR1, NANOG and several other proteins (Supplementary Fig. 6i, Supplementary Data File 3). Therefore, the landscape of pathogenic mutations impacting phase separation linking these mutations to phase dysregulation may provide an additional framework for understanding the pathological basis of diseases.

To determine which type of mutation is more relevant to phase separation, we calculated the frequencies of different mutation categories in disease-related phase-separating proteins, and found that pathological mutations of GP (5925, 17.6%) to hydrophobic amino

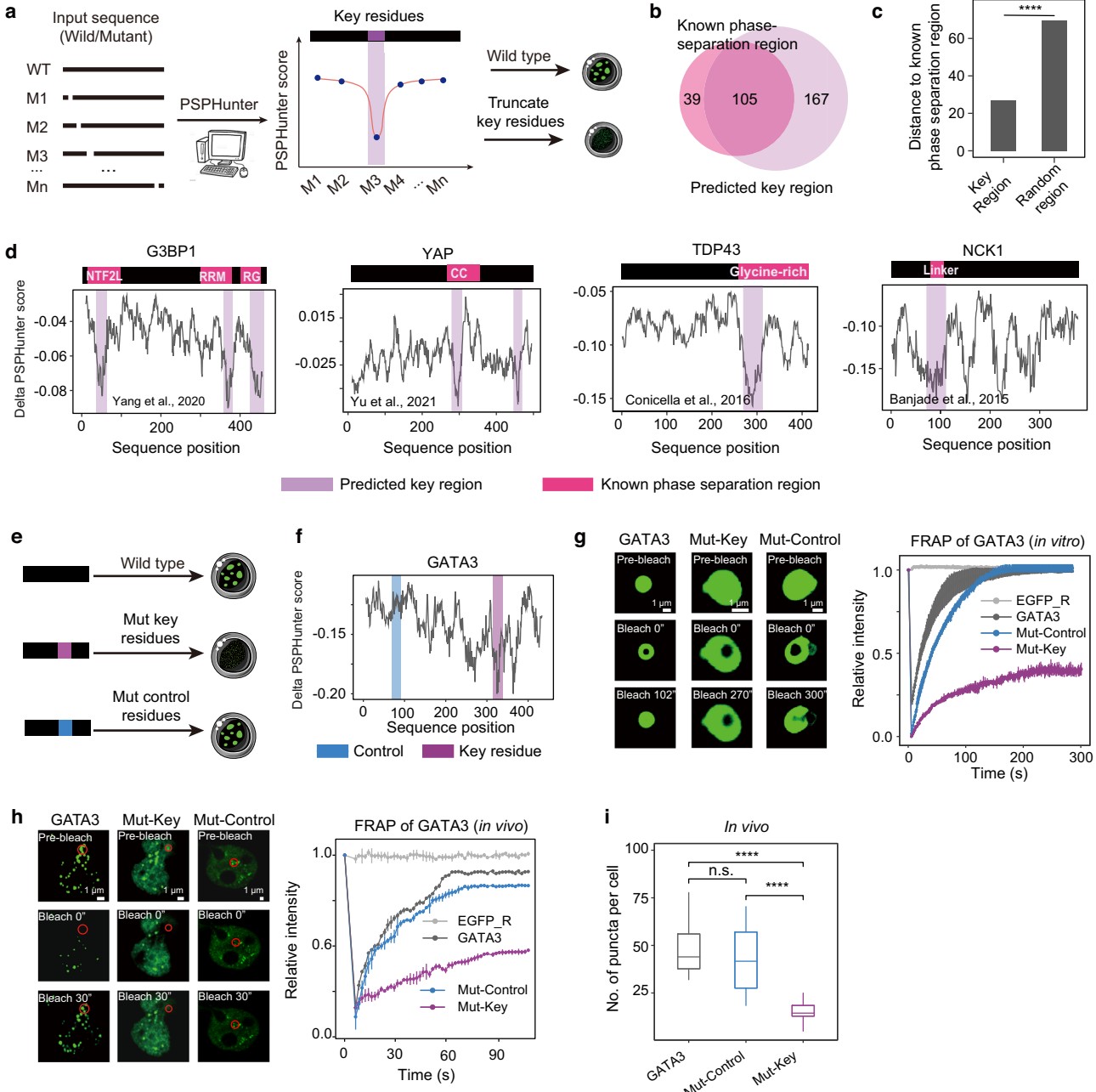

**Fig. 3 | PSPHunter can accurately predict key residues of phase-separating proteins. a** Strategy for identifying key residues. Mutation impact on phase separation is measured by changes in probability upon truncating specific units. **b** Venn diagrams showcasing the overlap between known phase-separation regions sourced and benchmarked from the PhaSePro database (pink) and the key regions predicted by PSPHunter (purple). **c** Distance comparison between the key region and an equal-length random region to the known phase-separating related region (one-sided Wilcoxon test, ****$P < 0.0001$, the bar denotes the average distance, Key region = 121; Random region = 121,000). Key regions are either within or near known regions. **d** Detailed comparison between known phase separation related region and key region predicted by PSPHunter. NTF2L, NTF2-like; RG, arginine-glycine rich; RRM, RNA recognition motif, CC, coiled-coil region; linker, inker between the first two SH3 domains. **e** Schematic representation of key residue validation, where the purple region represents the predicted key residue, expected to impact phase separation, while the blue region denotes the control residue with minimal effect on phase separation. **f** Identification of key residues and non-key

residues for GATA3 as defined by PSPHunter. Specifically, the key residues for GATA3 are located at amino acid positions 322–327, while the non-key residues are located at positions 88–93. **g–h** FRAP analysis of GATA3-GFP. Representative imaging (left) and GFP fluorescence intensity curve (right) demonstrate that the droplets formed by wild-type GATA3 and control residue-truncated GATA3 rapidly recover from photobleaching, whereas the droplets formed by key residue-truncated GATA3 exhibit limited recovery, $n = 3$, the error bars represent the standard deviation. **i** Quantification of puncta numbers for GATA3 and its mutants. The number of puncta of GATA3 per cell indicates that truncation of control residues has no significant reduction in puncta, whereas truncation of key residues significantly decreases. The boxplots were drawn from the lower quartile (Q1) to the upper quartile (Q3), with the middle line denoting the median, whiskers with a maximum 1.5 interquartile range (IQR), and outliers did not indicate the number of puncta (one-sided Wilcoxon test, ****$P < 0.0001$, not significant, denoted as n.s., $n = 3$).

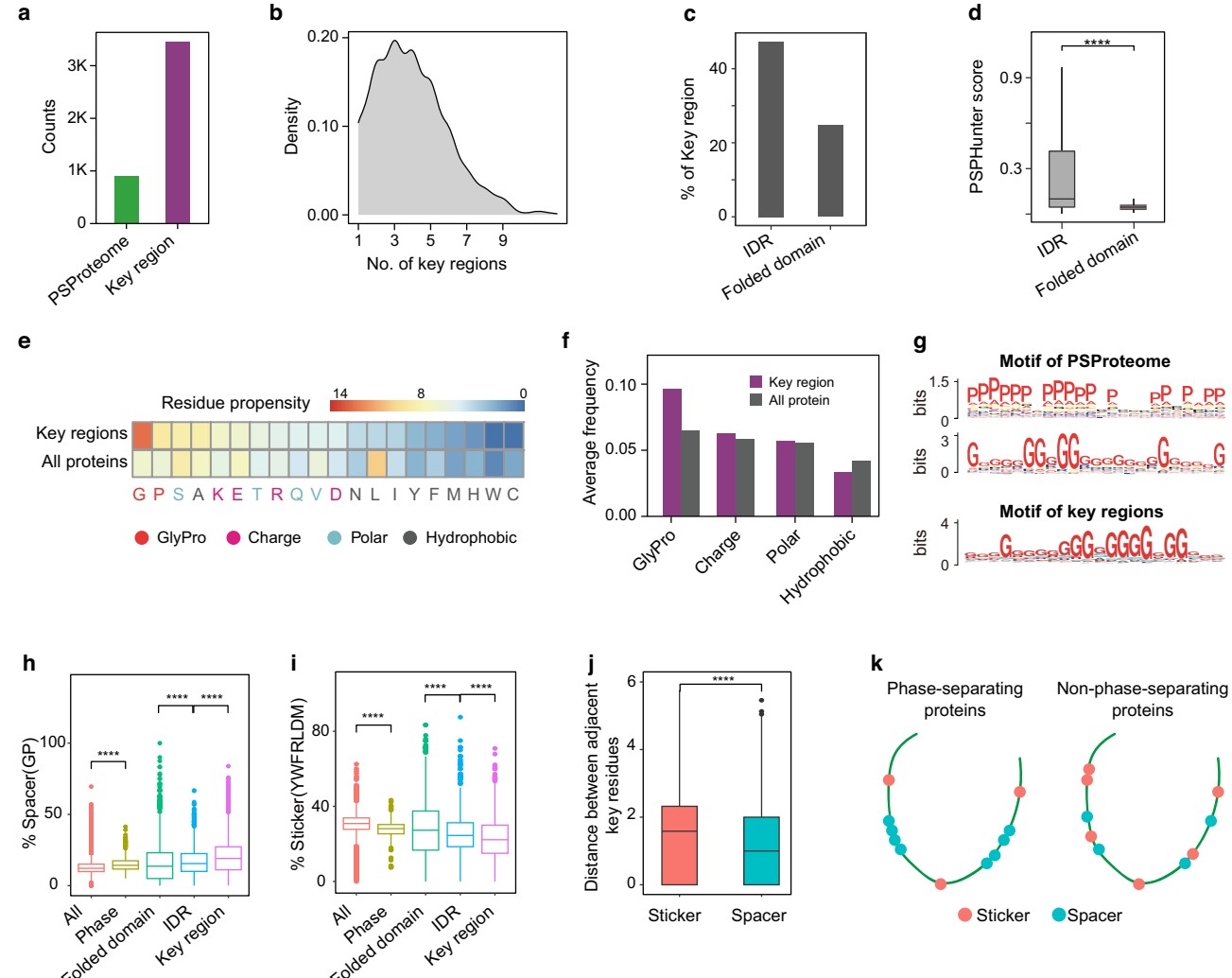

**Fig. 4 | Glycine and its motifs are enriched in spacer and key residues. a** Total counts of phase-separating proteins and their corresponding key regions. **b** Distribution of the number of the key regions shows that most phase-separating proteins have three or four key regions. **c** Overlap between key region and IDR region, RNA binding region and DNA binding region. **d** PSPHunter score of IDR region and folded domain (one-sided Wilcoxon test, ****P < 0.0001, Folded domain = 3395, IDR = 3450). **e–f** Residue propensity shows that key regions located in different areas tend to be GP-rich (red denoted glycine and proline, pink denoted charged residues such as DREK, cyan denoted polar residues such as VQTS, the remaining residues in grey are hydrophobic residues). **g** Representative motifs of phase-separating proteins and key regions. **h** Proportions of spacer amino acids in different protein types and sequence regions: Folded domain (representing nucleic acid binding regions predicted by SNBRFinder), IDR

(Intrinsically disordered region), and key region. The number of All = 20,420, Phase = 889, Folded domain = 4557, IDR = 2044, Key region = 3459. **i** Proportions of sticker amino acids in different protein types and sequence regions. Same number as **h**. **j**, Sequence distances of specific amino acid types in key amino acids. The number of Sticke = 18,869, Spacer = 17,439. **k** Model showing spacer residues GP tends to exhibit a contiguous pattern, while sticker residues prefer a dispersed distribution within phase-separating proteins. Note: All statistical tests were one-sided Wilcoxon tests. Significance levels are indicated by asterisks: *, P < 0.05; **, P < 0.01; ***, P < 0.001; ****, P < 0.0001 (not significant, denoted as n.s.). The box-plots were drawn from the lower quartile (Q1) to the upper quartile (Q3), with the middle line denoting the median, and whiskers with a maximum 1.5 interquartile range (IQR).

acids were more frequent compared to neutral mutations (Fig. 5g). This strongly suggested that mutations in GP residues were the most frequent in phase-separating proteins. In addition, mutations in GP residues were more likely to locate at key regions (Fig. 5h). Compared to the frequency of mutations in the random regions, GP mutations were preferentially located in the key regions (Fig. 5i and Supplementary Fig. 6j), and 80% of them had a significant impact on phase separation compared to other mutations regardless of location (Fig. 5j, k, Supplementary Fig. 6k,l, and Supplementary Data File 3).

Taken together, we concluded that pathogenic mutations, specifically those within key residues, impact phase separation capacity, particularly in the case of GP mutations (Fig. 5l). This may provide insights into the role of dysregulated phase separation in the development of diseases.

## Deletion of key residues disrupt the phase separation of GATA3 and promotes the migration and suppresses the growth of tumor cells

To investigate whether mutations in key residues affect the function of cells, we systematically analyze the pathogenic capacity of GATA3 defined by PolyPhen-2[70], a sequence-based method used to predict the functional effects of mutations. It shows that the key residues of GATA3 tend to possess higher pathogenicity and mutation frequency (Fig. 6a, b). Most of these high-frequent mutations are involved in breast cancer (Fig. 6c), strongly suggesting that they are closely associated with the phase separation of GATA3.

We chose the typical breast cancer cell line MCF7 to conduct further validations. The endogenous GATA3 was first knocked down based on the Tet-on system (Supplementary Fig. 6o, r). Then these

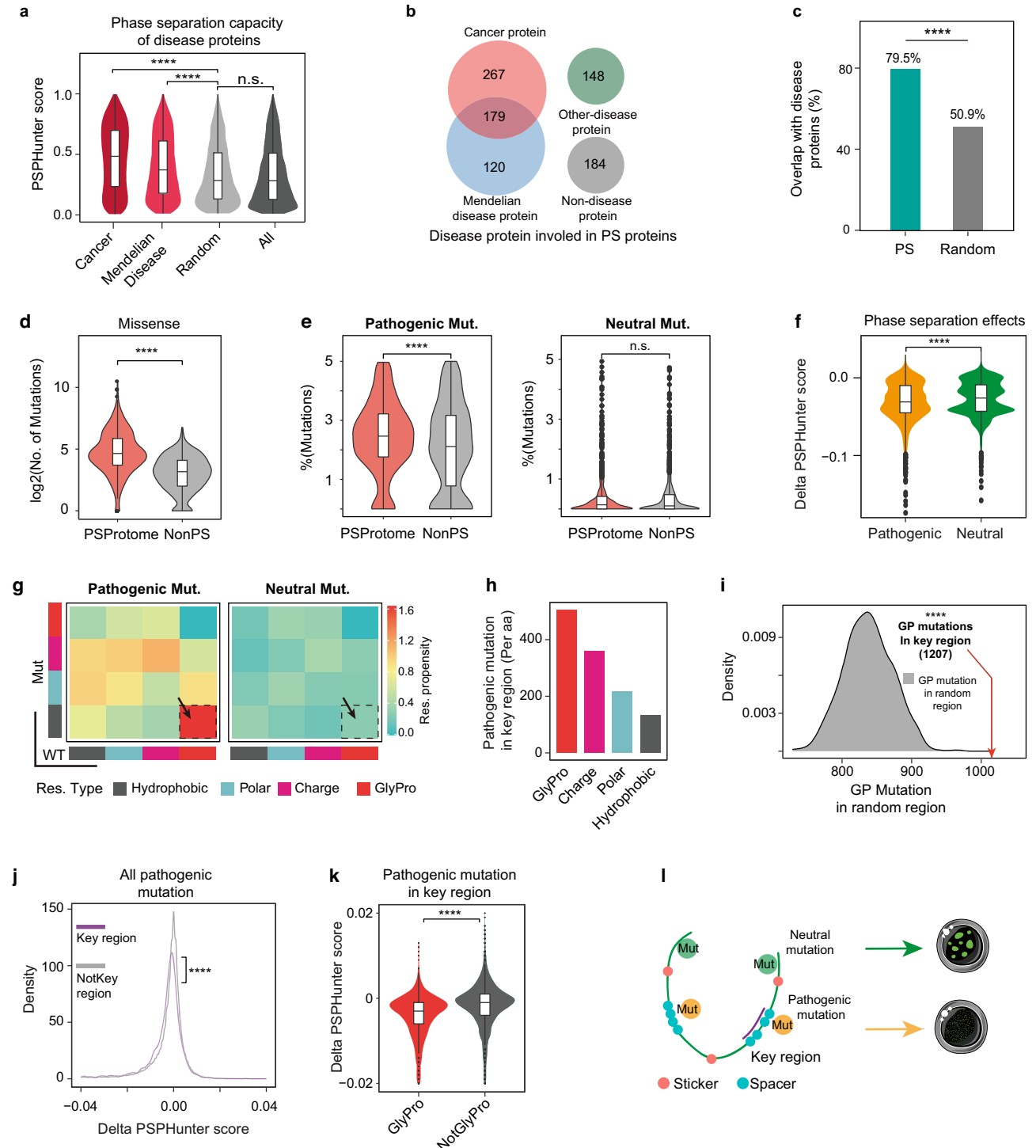

cells were transfected with PiggyBac GATA3, control residues truncated-GATA3 (Mut-Control) or key residues truncated-GATA3 (Mut-Key) for 48 h. Ectopic expression of these proteins in endogenous GATA3-knockdowned cells was confirmed by Western Blot (Supplementary Fig. 6p, q). To create a phase separation-rescued variant of GATA3, we engineered a fusion of the key residues-truncated GATA3 with the FUS IDR (referred to Mut+IDR). Upon expression of this fusion construct, a clear recovery of phase separation in GATA3 was observed (Supplementary Fig. 6m). Furthermore, fluorescence recovery after photobleaching experiments indicated that the restored puncta exhibited liquid-like properties (Supplementary Fig. 6n).

In the CCK8 assay, overexpression of Mut-Key significantly suppressed cell growth compared to the control ($P < 0.05$; Fig. 6d). Additionally, scratch healing assays revealed a marked increase in wound closure in key residue-truncated cells compared to control cells (Fig. 6e, f). Cell cycle analysis further confirmed this suppressive effect ($P < 0.01$; Fig. 6g, h), showing decreased proportions of cells in S and G2 phases alongside elevated G1 phase cells.

Notably, the G1 to S phase transition significantly regulates cell proliferation[71]. Intriguingly, the Mut+IDR group exhibited a distinct restoration in the proportions of cells in S phase (Fig. 6g, h). This finding suggests a potential role for GATA3's phase separation in modulating G1/S transitions to control cell proliferation. The rescue

**Fig. 5 | The pathogenic mutations glycine and proline disrupt phase separation more significantly than other mutations. a** Proteins associated with cancer and Mendelian diseases tend to have higher PSPHunter scores (one-sided Wilcoxon test, ****$P < 0.0001$, Cancer = 3999, Mendelian disease = 4498, Random = 3000, All = 20,150). **b** Overlap between phase-separating proteins and different types of diseases. **c** Nearly 80% of phase-separating proteins are disease-related (one-sided Student's $t$-test, ***$P < 0.001$; Random, $n = 1000$). **d** Phase-separating proteins have significantly more missense mutations (one-sided Wilcoxon test, ****$P < 0.0001$; PSProteome = 871; NonPS = 801;). **e** Phase-separating proteins have significantly more pathogenic mutations compared to the NonPS, with no differences in neutral mutations (one-sided Wilcoxon test, n.s. no significance, ****$P < 0.0001$; PSProteome = 891; NonPS = 891). **f** Pathogenic mutations have more impact on phase separation capacity than neutral mutations (one-sided Wilcoxon test, ****$P < 0.0001$; pathogenic = 684; neutral = 2684). **g** Heatmap showing that the mutations from GP to hydrophobic are the most frequent mutations in phase-

separating proteins. **h** Bar plot showing that GP is more likely to locate at key region (average mutations per residue). **i** Compare to the random regions which have the same length as the corresponding key region, the pathogenic mutations of GP are preferentially located at the key residues. **j** Boxplot showing that mutations in the key region have more impact on protein phase separation capacity (one-sided Wilcoxon test, ****$P < 0.0001$; KeyRegion = 4737; NotKeyRegion = 28,992). **k** Pathogenic mutations of GP in key residues have more impact on phase separation capacity than other mutations also within key residues (one-sided Wilcoxon test, ****$P < 0.0001$; GlyPro = 1207; NotGlyPro = 3530). **l** Model showing that pathogenic mutations of GP occurred in key region are more deleterious to protein phase separation capacity. Note: The boxplots depict the interquartile range (IQR) from the lower quartile (Q1) to the upper quartile (Q3). The median is indicated by the middle line within the box and Whiskers extend up to 1.5 times the IQR from the box.

experiments utilizing CCK8 assays and scratch healing assays collectively demonstrate that the restoration of phase separation capabilities can rescue the corresponding tumor cell phenotype (Fig. 6d–f). This substantiates the influential role of GATA3's phase separation in tumor cell growth.

Collectively, we showed that deleting key residues disrupts GATA3's phase separation and affects tumor cell function. Linking pathogenic mutation with phase separation might provide insight into disease research.

## Webservices provide tools for prediction of phase-separating proteins, key residues and pathogenic mutations impacting phase separation

PSPHunter offers resources of the phase-separating proteome, the landscape of phase-separating key residues and the landscape of phase separating-dysfunction pathogenic mutations, which will help to elucidate the functions of phase-separating proteins and explore the mechanisms of phase separation in transcriptional control, cell fate transition and disease development.

We also developed a user-friendly website (http://psphunter.stemcellding.org/, Fig. 7a) to implement PSPHunter using PHP, Perl-CGI, JpGraph, and Jmoe. We provide three modules using only protein sequences as the input to identify putative key residues, predict protein phase separation capacity and evaluate the phase separation effect of mutations (Fig. 7b).

## Discussion

We have developed PSPHunter, a multi-information fusion-based machine learning model for predicting phase-separating proteins and the residues key phase separation to promote the application and biological discovery. Compared to the first-generation phase-separating protein predictors[72] that were based on small samples and specific features, PSPHunter harnessed a diverse range of features, including sequence embeddings, protein evolutionary attributes, functional features like protein-protein interaction network topology, and post-translational modifications. This integration of sequence and functional features led to improved performance, highlighting their complementary nature.

Furthermore, of greater significance, we extended the applicability of PSPHunter from protein-level to residue-level analysis by introducing a sliding window scanning strategy to identify the residues with the most impact on phase separation. PSPHunter can minimize the size of fragment that can impact phase separation to six or fewer residues as opposed to the longer IDRs or prion-like domain. Applying PSPHunter for the identification of key residues within the PSProteome, we observed a significant enrichment of glycine (G) and proline (P) among these key residues, along with the presence of spacer-sticker patterns. PSPHunter, being an AI model, introduces a more diverse range of features to describe the attributes of phase-

separating proteins, potentially encoding a broader spectrum of mechanisms.

In addition, PSPHunter can also evaluate the influence of mutation on phase separation, and provide a mechanistic link between phase separation and pathological mutations. Thus, a map of key residues that impact phase separation and are associated with disease-related mutations can be a powerful tool to investigate the molecular mechanisms underlying these diseases. We also observed a preference for GP mutations within key residues, which had a notable impact on phase separation capacity. However, it is important to acknowledge that further experimental evidence is required to fully validate the conclusion that pathogenic mutations indeed affect phase separation.

Specifically, the word2vec descriptor is the closest to PSPHunter in terms of predicting the effectiveness of key residues (Supplementary Fig. 7a–f), and is based on the premise that the amino acids key phase-separation may follow a specific syntax. Characterization of the key residues reveals the diversity of protein phase separation mechanisms, which could provide insights into how mutations specifically affect phase separation and disease development, including the discovery of pathological mechanisms and therapeutic targets with potential clinical implications[73–78].

However, validation of predicted key residues is technologically limited to relatively small experimental sample sizes. Though the effect of pathological mutations on phase separation has been globally evaluated, we have not established a direct link between the observed phase separation dysregulation and disease phenotype. The single mutation usually has limited effect on the capacity of phase separation, which may obfuscate key mutations in specific proteins. Expansion of phase-separating protein resources, refined feature characterization, as well as detailed molecular studies of specific proteins in disease-appropriate model systems may help overcome these limitations in the future.

## Methods
### Datasets of phase-separating proteins

Given the scarcity of resources on phase-separating (PS) proteins and the complexity in defining phase separation, we aimed to gather data comprehensively. Initially, we collected phase-separating proteins from various sources, including PhaSepDB[37], LLPSDB[38], DrLLPS[39] and PhaSePro[40] (Supplementary Fig. 1a, Supplementary Data File 4). We manually retrieved scaffold proteins pivotal in phase separation, serving as primary components undergoing liquid-liquid phase separation either independently or in conjunction with co-scaffolds. This aggregation yielded 488 phase-separating proteins across 47 species, referred to as MixPS488. Our secondary objective was twofold: to conduct methodological comparisons and to gauge the impact of dataset size. From PhaSepDB. we collected 237 PS-Self proteins (proteins that can undergo self-assembling PS in vitro) spanning 42 species, termed MixPS237. Additionally, we specifically curated a dataset

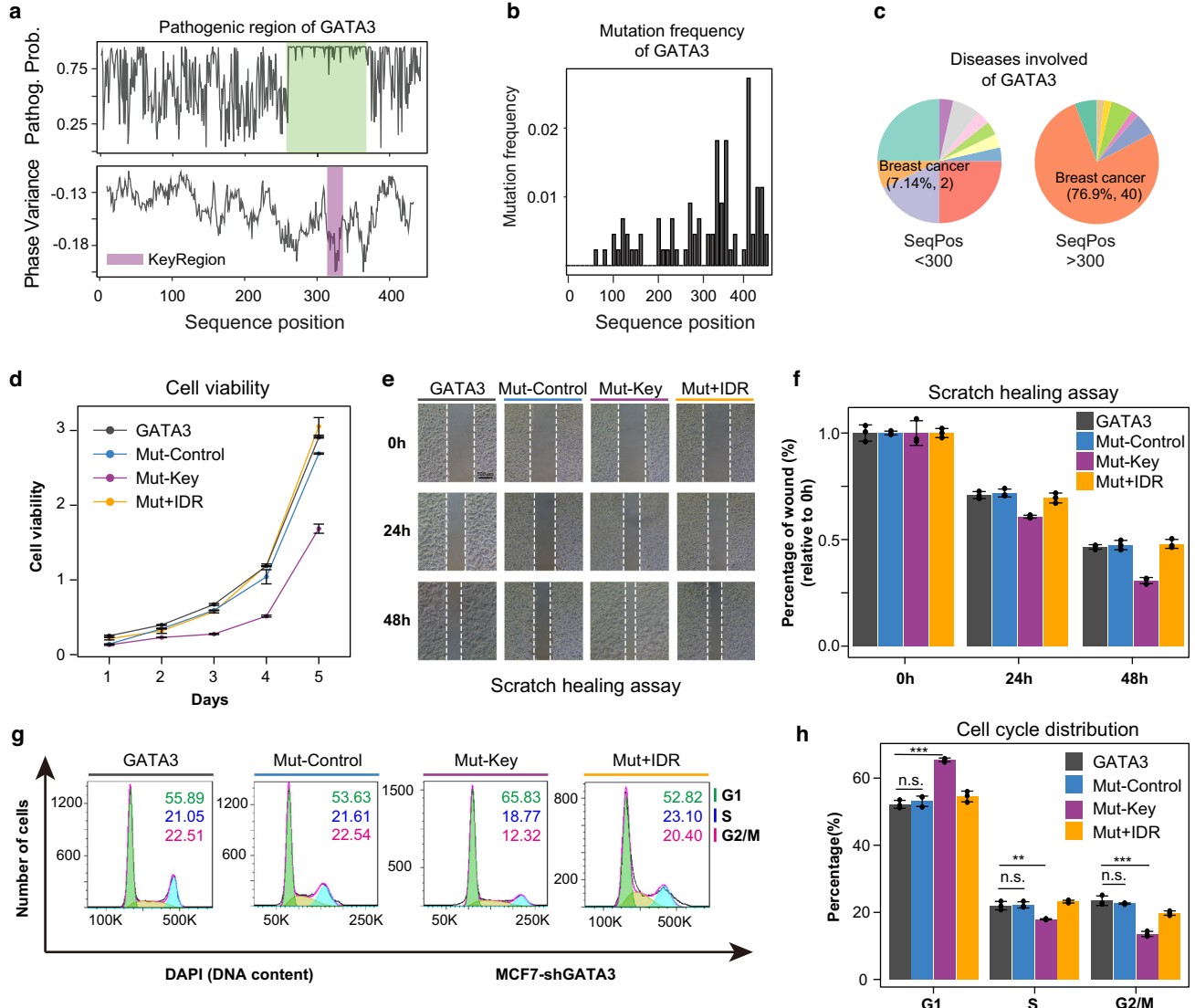

**Fig. 6 | Deletion of key residues disrupt the phase separation of GATA3 and promotes the migration and suppresses the growth of tumor cells.**
**a** Pathogenic probability of GATA3 defined by PolyPhen2, a sequence-based method used to predict the functional effects of mutations. It shows that the key regions of GATA3 tend to possess higher pathogenicity. **b** Bar plot showing that most of the mutations occur in the end of the GATA3. **c** Most of the high-frequent mutations are involved in breast cancer. **d** Cell viability was reduced by over-expression of key residues truncated-GATA3 in MCF7 cells ($n$ = 3 biologically independent experiments, same as **f**, **h**). Line in purple denotes overexpression key residues truncated GATA3, line in blue denotes overexpression the control residues truncated GATA3, line in yellow denotes the the overexpression of the phase separation-rescued variant of GATA3, and line in grey denotes overexpression the wild type GATA3. **e**, **f** Effect of GATA3 on MCF7 cell migration. Representative result of scratch healing assay (**e**) and statistical analysis of scratch healing assay (**f**). **g**, **h** Illustrative outcomes of cell cycle analysis (**g**) and the corresponding distribution (**h**) of MCF7 cells across the G1, S, and G2/M phases. Note: All statistical tests were one-sided Wilcoxon tests. Significance levels are indicated by asterisks: *$P < 0.05$; **$P < 0.01$; ***$P < 0.001$; ****$P < 0.0001$ (not significant, denoted as n.s.). The error bars represent the standard deviation.

focused on human-specific phase-separating proteins. Integrating the latest literature findings, we constructed hPS167, comprising 167 human phase separation scaffold proteins.

## Datasets of non-phase-separating proteins
Given our collection of PS proteins from 47 organisms, we assembled corresponding proteomes from the Swiss-Prot database to serve as background proteins. To minimize data redundancy, we specifically selected 7 organisms with at least 10 recorded instances of PS proteins in our datasets (Homo sapiens, Saccharomyces cerevisiae, Drosophila melanogaster, Caenorhabditis elegans, Mus musculus, Xenopus laevis, Schizosaccharomyces pombe). Subsequently, protein sequences documented in MixPS488, MixPS237, and hPS167 datasets were excluded. As phase separation is deemed to be driven

by multivalent interactions between multiple folded domains or disordered domains, we extracted single-domain proteins from background proteins as negative candidates, utilizing PfamScan[79] against the Pfam-A database. The remaining proteins underwent blastclust algorithm[80] analysis with a sequence identity threshold set at 0.3 to diminish sequence similarity. Ultimately, 16851 proteins, meeting the quality control criteria, constituted the non-phase-separating (non-PS16851) protein set.

Additionally, we developed a human-specific dataset of non-phase-separating proteins by using UniProt[81] database keywords 'human' and filter criterion 'Reviewed', excluding hPS167. Further refining these single-domain proteins based on a pairwise sequence identity of less than 30%, yielded a subset of 5754 proteins (non-hPS5754).

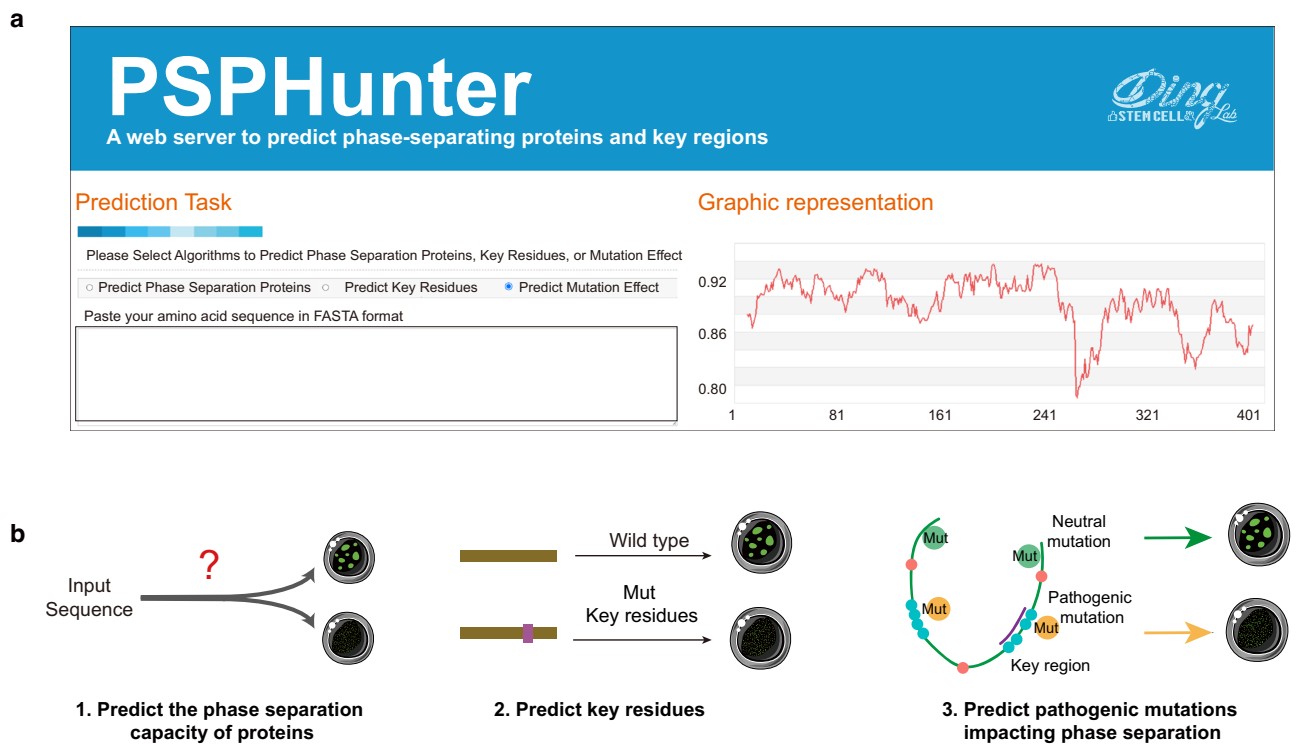

**Fig. 7 | Webservices provide tools for prediction of phase-separating proteins, key residues and impact of pathogenic mutations on phase separation. a** Homepage of PSPHunter. **b** Function of PSPHunter services. PSPHunter provides features to identify putative key residues, predict protein phase separation ability and evaluate the phase separation effect of mutations.

## Construction of training and testing datasets

The phase-separating proteins within the Mix-PS488, Mix-PS237, and hPS167 datasets underwent further refinement by limiting sequence similarity between any two proteins to below 30%, resulting in 379, 172, and 135 positive samples, respectively. To generate negative samples, an equal number of proteins were randomly selected from non-PS16851 and non-hPS5754 to ensure species specificity. This iterative process was repeated 100 times. Subsequently, the paired positive and negative samples were divided into training (70%) and independent testing (30%) datasets, resulting in 100 training datasets and 100 testing datasets. This procedure led to the creation of Training datasets for Training_I_MixPS488, Training_II_MixPS237, and Training_III_hPS167. Correspondingly, the independent test datasets comprised Independent_Test_I_MixPS488, Independent_Test_II_-MixPS237, and Independent_Test_III_hPS167. Additionally, human proteins were excluded from the MixPS488 and MixPS237 datasets to construct a non-human test set, namely Independent_Test_IV_NonHuman_MixPS237 and Independent_Test_V_NonHuman_MixPS488. The final evaluation on the independent test dataset represents the average performance across these 100 sets of data.

## Overview of the PSPHunter algorithm

In our previous works, we developed a series of algorithms to predict functionally important residues by combining machine learning- and template-based strategies[82,83]. Here, this framework was extended to the prediction of phase-separating proteins and key residues. To construct PSPHunter (Fig. 1a), we characterized each protein using 4 groups of sequence-based descriptors including amino acid composition, evolutionary conservation, predicted functional site annotations, and word embedding vectors. Additionally, we implemented 2 groups of functional features including protein annotation information and network properties. Based on the complementarity of sequence attributes and functional attributes, we constructed a machine learning-based model for phase-separating protein prediction. In addition, by elaborating on the sensitivity of word2vec feature to sequence changes in response to phase separation variations, we aim to convey that the feature of word2vec can capture subtle alterations in the protein sequence, which may correspond to key residues responsible for key the phase separation.

## Amino acid composition

According to Quiroz's research about how intrinsically disordered protein encoded phase behavior[84–86], we classified each amino acid into categories of polar amino acids (N, Q, S, T), charged amino acids (R, K, D, E), hydrophobic amino acids (L, A, V, I, F, Y, M, H, W, C). G and P are placed into a separate category given their unusual structure in triggering phase separation. In this study, we calculated two features to represent residue composition. The first feature is the percentage of all residues in each category to the total number of amino acids. The second feature is the percentage of the consecutive two amino acids. Specifically, every two consecutive amino acids are regarded as a unit and the category information was further assigned to the unit. Then the percentages of doublet categories such as polar-polar, polar-charge were calculated to represent the second residue composition feature. Collectively, we generated a 20-dimensional vector to represent the residue composition feature.

## Evolutionary conservation

Evolutionary conservation features consist of three parts: PSSM, distribution of residue conservation score, and HMM profile. PSSM, also known as Position-Specific Scoring Matrices, is a sequence profile comprising the evolutionary information of a protein sequence. In order to construct this profile, a search against the NR (Non-Redundant) database from NCBI was conducted for each query, employing the PSI-BLAST program[87] (v 2.2.31+) with the parameters ($j = 3$ and $e = 0.001$). By following this procedure, a substitution frequency

matrix was obtained, representing the likelihood of amino acid substitutions at each position among the 20 different amino acid types.

To characterize each protein, a Z-score transformation was initially applied to each row of the matrix. Subsequently, for each column, the average score for each amino acid was computed. As a result, the entire matrix was condensed into a 20-dimensional vector that encapsulates the evolutionary properties of each residue type. This vector was then utilized to summarize the average scores based on the categories of polar, charged, hydrophobic, and GP residues. Ultimately, to represent each protein, a concise 4-dimensional vector was generated, which encapsulates the summarized information from the aforementioned categories.

In addition to PSSM, we implemented relative entropy[88] generated based on the output by PSI-BLAST. Each residue of the query sequence obtained a conservation score that could characteristic its evolutionary signature. To represent the distribution of conservation score for each protein, we calculated 5 statistical characteristics including maximum, minimum, first quartile, second quartile and third quartile. The 5-dimensional vector was therefore used to represent the residue conservation score.

The HMM profile, specifically referred to as a profile hidden Markov model (HMM), is a powerful tool for elucidating the distant homologous relationships that exist among proteins. In our study, we adopted the HHblits[89] program (v 2.0.16) with default parameters to generate the HMM profile. This program facilitated a comprehensive search of the query protein against the UniProt20 database. Similar to the generation of the PSSM profile, the HMM profile of each protein was first Z-score normalized by row and then averaged by column, which would result in a 20-dimensional vector in response to each residue. This 20-dimensional vector was further condensed to 4 by summing the average scores according to residue categories of polar, charge, hydrophobic and GP.

## Predicted functional site annotations

Sequence-derived functional features can be used to reflect the functional preference of a protein, and the intrinsically disorder region and protein post-modification had been approved in relation to phase separation. In the current study, the secondary structure states of residues and the accessible residues with relative solvent accessibility greater than 20% were assigned by the SPIDER2 program[90]. The disordered residues were identified using the SPINE-D program[91] (v 2.0.0). RNA and DNA binding residues were predicted by SNBRfinder[92]. For each sequence, we computed the percentages of helix residues, sheet residues, coil residues, accessible residues, disordered residues, RNA binding residues and DNA binding residues. We adopted GPS[93–96], a comprehensive PTM predictor, to extract the all the potential phosphorylation site, methylation site, S-nitrosylation site and palmitoylation site of each protein. Regarding to the predicted mutation information, we utilized Rhapsody[97](v 1.0) to conduct a saturation mutagenesis of each residue. According to the output of Rhapsody, we computed the percentages of deleterious and neutral mutations. In addition, we implemented the protein length to represent the size of protein. Finally, each protein was represented by a 14-dimensional vector.

## Word embedding vectors

Word embedding is a technique widely used in natural language processing and has been applied to computational biology[98–100]. In this study, we leveraged word embedding to analyze protein sequences. To facilitate this analysis, we adopted the word2vec method[101], which is a popular approach for constructing the distributed representations. To apply word embedding to protein sequences, we treated each protein sequence as a sentence, and its subsequences were considered as individual words. By doing so, we aimed to capture the inherent structure and context within the sequences. The distributed representations were then constructed using the word2vec method, which enabled us to generate numerical vectors that encode the semantic properties of each word (subsequence) in the protein sequences.

Generally, training a word embedding model typically requires a corpus, which serves as the basis for capturing the relationships between words. In our research, we selected the corpus by testing two datasets, namely PS135 and SwissProt, which provided a diverse range of protein sequences for analysis. Following the recommendation of previous studies[102], each protein sequence was transformed into three sequences that were composed of nonoverlapping 3-grams. To implement the word2vec method, we adopted the Gensim package (https://radimrehurek.com/gensim/) and employed a bag-of-words model. Several parameters were set to ensure optimal performance. The maximum distance between the current and predicted words was set to 70, while the dimensionality of word vectors was set to 60 based on the parameter selection using the PS135 dataset (Supplementary Fig. 1f). Consequently, each protein was represented by a 60-dimensional vector.

## Protein annotation information

In this section, we utilized the experimental validated PTM and mutation information to annotate each protein. We downloaded four types of post-modification information including phosphorylation, acetylation, ubiquitination and methylation from PhosphoSitePlus[103] database. The PTM frequency of each modification was defined as the number of annotated sites divided by the protein sequence length. We extracted the mutation information from the HuVarBase[69] database which is a comprehensive database integrating resources of 1000 Genomes, ClinVar, COSMIC, Humsavar and SwissVar. By searching with uniport ID of each protein, we finally obtained 774,863 variants from 18,318 proteins. Among the variants 702,048 are disease causing and 72,815 are neutral variants. The protein expression abundance information was extracted from PAXdb[104] (2017, organ: WHOLE_-ORGANISM). The protein age of each query was inferred using phylogenetic analysis and extracted from ProteinHistorian, in which we selected 'PPODv4_Jaccard_families' as the protein family database and 'Wagner parsimony' as the ancestral reconstruction algorithm[105]. The essential gene list including 1,216 genes is the consensus results generated by the genome-wide single-guide RNA screening[106] and the haploid gene-trap screening[107]. The housekeeping gene list consists of 8,874 genes expressed in all tissues[108]. Collectively, each protein was denoted by a 11-dimensional vector from an evolutionary perspective.

## Network properties

Because phase-separating proteins generally have similar biological functions, these proteins would be expected to be densely located in protein-protein interaction (PPI) networks. Based on this assumption, we established networks by using the PPIs from the HIPPIE database[109]. Based on the complete PPI networks, we calculated four generic properties, which comprised degree, betweenness, clustering coefficient, average neighbor degree.

## Classification model

After extracting the above features, we developed models to predict phase-separating proteins. In this work, we evaluated six types of ma Rhapsodychine learning algorithms, including support vector machine (SVM), naïve Bayesian classifier (NB), neural network (NN), random forest (RF), light gradient boosting machine (LightGBM), and extreme gradient boosting (XGBoost). All these algorithms were implemented using the scikit-learn package[110] (v 1.2.0). The parameters c and g of the radial basis function were set to 2 and 0.125, respectively, in SVM, and the number of trees was set to 500 in RF. Other parameters were set to the default in the implementation process. To establish the final model, we integrated the multifaceted features by testing three types

of ensemble strategies, including the average of the probability scores from the first layer, the direct integration of different features into a single model, and the two-layer stacking model. By comparing the performance of different ensemble strategies (Supplementary Fig. 2a–c), the direct integration model based on random forest was finally selected.

To evaluate the predictive performance, 5-fold cross-validation was conducted on the primary dataset. The target dataset was initially divided into 5 subsets, each containing an equal number of proteins. During each cross-validation iteration, one subset was employed as the test set, while the remaining subsets served as the training set. This process was repeated 5 times, with each subset serving as the test set once, enabling the calculation of the average performance.

### Evaluation indicators

The primary measure of prediction performance was assessed using the area under the receiver operating characteristic curve (AUC). The AUC quantifies the classifier's ability to distinguish between true positives and false positives at various classification thresholds. Additionally, other well-established metrics, including recall, precision, F1-score, accuracy (ACC), and Matthews correlation coefficient (MCC), were computed as follows: TP, TN, FP, and FN represent the numbers of true positives, true negatives, false positives, and false negatives, respectively.

$$Recall = \frac{TP}{TP + FN} \tag{1}$$

$$Precision = \frac{TP}{TP + FP} \tag{2}$$

$$F1 - score = \frac{2 \times Recall \times Precision}{Recall + Precision} \tag{3}$$

$$Accuracy = \frac{TP + TN}{TP + FN + TN + FP} \tag{4}$$

$$MCC = \frac{TP \times TN - FP \times FN}{\sqrt{(TP + FN)(TP + FP)(TN + FP)(TN + FN)}} \tag{5}$$

### Feature importance and feature selection

Feature importances were computed using the fitted 'feature_importances_' attribute from the scikit-learn package[110]. We performed feature selection on all 123 sequence and functional features. These features were then ranked based on their importance value, and we assessed their contribution by progressively increasing the number of features. We observed that the model's performance reached its best when the number of features reached 60, with no significant improvements upon further increase. Consequently, we settled on 60 features for the final model.

### Key residue detection

In this study, we employed three distinct strategies for screening these key residues. Specifically, the first strategy is designing features to make the PSPHunter sensitive to amino acid variations in the sequence. Initially, we incorporate word2vec features, which encode the combination patterns of short sequence fragments. Furthermore, we compress and integrate residue-level features such as PSSM and HMM, enabling the transmission of sequence variations at the amino acid level to the protein scale. These residue-level descriptions enhance the sensitivity of PSPHunter in detecting sequence variations that are associated with the protein's phase separation capacity. The second strategy aimed to study the influence of each amino acid

on the capability of the protein to undergo phase separation. We achieved this by treating 20 consecutive amino acids as a unit to represent a given residue at its middle position (Fig. 3a, Supplementary Fig. 4a). We selected a truncated-unit value of 20 amino acids based on the average length of phase-separation proteins (~600 amino acids). This bin size is 1/30, which we consider relatively reasonable. It is worth noting that theoretically, a larger value for this parameter leads to a greater variation in the delta PSPHunter score. Users have the flexibility to adjust this parameter in our standalone version.

Subsequently, we employed PSPHunter to calculate the phase-separation probability for each unit-deleted protein. After evaluating the effect of all truncation possibilities, we can obtain a curve of the effect of each residue (excluding 10 residues at each end) in response to each truncation. The residues with the greater deviation from the average phase separation capacity are considered as the key residues.

### Key region detection

The consecutive key residues are treated as key region (Supplementary Fig. 4a). To balance PSPHunter's sensitivity in identifying key residues, we aim to capture consecutive amino acids with the greatest impact on phase separation. To achieve this, we have set an empirical parameter, selecting 20-40 top-ranked key residues as candidates, aligning with 1-2 times the truncated unit. This balances user convenience and prioritizes crucial regions. We next connect consecutive amino acids in candidates to form key regions. For instance, we kept the top 1% of the total number of residues when the sequence length is greater than 2000, 2% when the sequence length is between 1000-2000, 4% when the sequence length is between 500-1000, and 5% when the sequence length is less than 500. After that, the retained residues are attached according to the sequence position which would further form to key regions.

### Other phase separation predictors

PLAAC[111], LARKS[112], R + Y[26], DDX4-like[28], catGRANULE[113], PScore[114] and CRAPome[115] were first-generation phase-separating protein predictors summarized by a recently review[72]. Youn et al.[42] further utilized these predictors to analysis the properties of stress granule and P-body proteomes. We obtained all the predicted probabilities of each protein from its original supplemental information. Additionally, the predicted results of PhaSePred[44] (http://predict.phasep.pro/), MAGS[116] (https://github.com/ekuec/2019_StressGranuleFeatures/) and PSPredictor[45] (http://www.pkumdl.cn/PSPredictor/) were obtained from the corresponding web server and GitHub repositories.

### Cell lines and culture conditions

The MCF7 cell line was a gift from Dr. Hai Hu (Sun Yat-sen Memorial Hospital). The HEK293T cell line was kindly gifted by Dr. Jianlong Wang from Icahn School of Medicine at Mount Sinai. HEK293T were grown in DMEM medium (Hyclone, SH30022.01) containing 10% FBS (LONSERA, S711-001S), and MCF7 were grown in RPMI 1640 (Gibco, C11875500BT) containing 10% FBS (LONSERA, S711-001S). All cells were cultured at 37 °C with 5% $CO_2$.

To establish exogenous expression of GATA3, Mut-Control, Mut-Key, and Mut+IDR in the MCF7 cell line, achieving levels of expression similar to endogenous GATA3 in MCF7, we first established an MCF7-shGATA3 cell line. Endogenous GATA3 expression in this cell line was silenced using short hairpin RNA (shRNA) delivered via doxycycline (DOX) treatment. The shRNA oligonucleotides were designed using the tool available at http://www.broadinstitute.org/rnai/public/gene/search. The forward primer sequence for shGATA3 is CTAGGCCAAGAAGTTTAAGGAATATCTCGAGATATTCCTTAAACTTCTTGGCTTTTTG, and the reverse primer sequence is AATTCAAAAAGCCAAGAAGTTTAAGGAATATCTCGAGATATTCCTTAAACTTCTTGGC.

## Motif analysis

Motifs of phase-separating proteins and key residues were discovered by the MEME 5.4.1 server[117] with motif size = 6–8 and other parameters at default.

## Functional enrichment analysis

To explore the functional roles of proteins involved in phase separation, we identified the associated GO terms using Metascape[118], in which 'Homo sapiens' were selected as the background and all the 898 phase separation proteome were chosen as the input. Then 'Custom Analysis' was adopted to further analysis. The over-represented GO biological processes, cellular components, molecular functions, and diseases types were reserved with default parameters.

## Protein expression and purification

cDNA encoding the interested proteins were cloned into pET28a expression vector. The base vector was engineered to include His-tag followed by EGFP. All expression constructs were sequenced to ensure sequence identity. For protein expression, plasmids were transformed into BL21 E.coli (TransGen Biotech, CD601-02) and grown as follows. A fresh bacterial colony was inoculated into LB media containing kanamycin and grown overnight at 37 °C. Then the cells were diluted 1:30 in 300 mL LB with freshly added kanamycin and grown at 37 °C for approximately 5 h to make sure OD600 up to 0.6–0.8. Then IPTG (Solarbio, I1020-5) was added to 0.3 mM and growth continued overnight at 16 °C for 18 h. Protein purification was performed according to the instruction of Protein Purification Kit (Cwbio, CW0894S). The recombinant EGFP fusion proteins were concentrated in Amicon Ultra 30KDa centrifugal filters (Millipore, UFC803024) for use.

## In vitro droplet formation

The recombinant EGFP fusion proteins were concentrated and desalted to an appropriate concentration using Amicon Ultra centrifugal filters (Millipore, UFC803024). And then the interested proteins were added to droplet formation buffer, which consists of 50 mM Tris-HCl pH 7.5 (Thermo Fisher, 15567-027), 10% glycerol (Sigma, G5516), 1 mM DTT (Sigma, D9163), 10% PEG8000 (Sigma, H-209Z-0T968) and 125 mM NaCl (Sigma, S5150). The protein solution was immediately loaded onto a homemade chamber, and then imaged with microscopy (Nikon Eclipse Ts2R-FL).

## Fluorescence recovery after photobleaching (FRAP)

HEK293T stably expressing recombinant EGFP fusion proteins were cultured in glass bottom dish for 24 h. Stable cell lines were obtained after drug selection. Fluorescence images of EGFP were acquired on a Nikon A1+ confocal microscope with 488 nm laser using a 100x oil-immersion objective lens (HP Apo TIRF 100xH, 1.49 NA, Nikon). The fluorescence intensity of bleached cell at each time point was normalized by fluorescence intensity at background region and the fluorescence intensity of the adjacent unbleached puncta. The images were analyzed using NIS-Elements software. The inner fluidity of in vitro droplets was also evaluated by FRAP.

## Puncta analysis

To quantify the puncta of in cells, cells were imaged by confocal microscopy using the same parameters across different groups (GFP-fused GATA3, control region truncated GFP-fused GATA3 and key region truncated GFP-fused GATA3). Using Imaris software (Bitplane), the number of puncta in cells was calculated with the spot module, respectively.

## Protein extraction and western blots

Proteins were extracted in Cytobuster (Merck) at room temperature for 10 min. The proteins were mixed in 5×SDS buffer (Bio-Rad) and were separated on 12% Bis-Tris gel at 100 V for 90 min, and then wet-transferred to a 0.45 um PVDF membrane (Millipore) in ice-cold transfer buffer in 300 mA for 2 h or in 40 V for 12 h. After blocked with 5% BSA in TBS for 1 h at room temperature with shaking, the membrane was incubated with the primary antibody overnight at 4 °C (anti-GATA3, Rabbit, Monoclonal, ABclonal, Cat. number A19636, Lot number 4000000115, Dilutions/amounts 1:1000; anti-β-Tubulin, Rabbit, Polvclonal, ABclonal, Cat. number AC008, Lot number 3523022349, Dilutions/amounts 1:200). After washed three times with TBST for 5 min at room temperature, the membrane was incubated with 1:1000 secondary antibodies for 1 h at room temperature. After washed three times with TBST for 5 min at room temperature, the membrane was developed with ECL substrate (Thermo Fisher) and imaged using a CCD camera.

## Reporting summary

Further information on research design is available in the Nature Portfolio Reporting Summary linked to this article.

## Data availability

The pre-established model, along with the associated training and testing datasets of PSPHunter, and the predicted phase separation probabilities of all human proteins using PSPHunter, are available at https://github.com/jsun9003/PSPHunter. Uncropped scans of all blots and gels in Figures, along with the relevant raw data from each figure or table, are provided in the Supplementary Information/Source Data file. Source data are provided with this paper.

## Code availability

The codes of PSPHunter developed in this study are available at https://gitsshub.com/jsun9003/PSPHunter, with DOI: 10.5281/zenodo.10791112.

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

## Acknowledgements
This research was funded by grants from the National Key R&D Program of China (2023YFA1800900), National Natural Science Foundation of China (31970811 and 32170798), the Guangdong Basic and Applied Basic Research Foundation (2021B1515120063) to J.D., the Natural Science Foundation of Guangdong Province, China (2021A1515010938, 2023A1515010148), the Fundamental Research Funds for the Central Universities, Sun Yat-sen University (23PTPY88) to J.S., the National Natural Youth Science Foundation of China (Grant No. 32100927) to H.Y., Post-doctoral Program (2021M703760) and National Natural Science Foundation of China (32100497) to C.W., National Natural Science Foundation of China (32270679) to J.W., National Natural Science Foundation of China (81970481) to Y.Y., Fundamental Research Funds for the National Natural Science Foundation of China (82304746), Guangdong Basic and Applied Basic Research Foundation (2021A1515111056) to L.F., 1.3.5 project for disciplines of excellence, West China Hospital, Sichuan University (ZYJC18010) to X.Z.

## Author contributions
J.D. and J.S. conceived and designed the study. J.S. conducted data analysis. J.Q., C.Z., X.L. and X.Z. performed experimental validations. J.S., J.Q. and C.Z. drafted the manuscript with input from all other authors. All authors contributed to result discussions and manuscript editing. Y.K. refined the web services. T.C. polished the manuscript. H.Y., Y.Y., Y.K., D.H. and L.F. provided support, while H.Y. and J.D. supervised the research.

## Competing interests
J.D., J.S., J.Q. and C.Z. are listed as inventors of a patent applications titled 'A machine learning method for predicting phase separation driving residues', the remaining authors declare no competing interests.
