## [Peer Review File · Nature Communications]

Reviewers' Comments:

Reviewer #1:

Remarks to the Author:

The authors have made great efforts in the revision and addressed the majority of my concerns. I am delighted to extend my support for the publication of their manuscript pending the addressing of the following minor points:

1. Authors should remove the term 'in vitro' in the legend of Fig.3i, as it is inconsistent with what is shown in the figure.
2. Fig.6e and f were not correctly referenced in the text 'Intriguingly, the Mut+IDR group exhibited a distinct restoration in the proportions of cells in S phase (Fig.6e.f).'

Reviewer #2:

Remarks to the Author:

The manuscript "Precise prediction of phase-separation key residues by machine learning reveals the dysregulated landscape impacted by pathogenic mutations" presents a study in the field of biophysics and molecular biology. The authors developed PSPHunter, a machine learning algorithm, to predict key residues involved in protein phase separation. This tool is validated both in vitro and in vivo, demonstrating its potential in identifying disease-associated phase-separating proteins and understanding the mechanisms governing transcriptional control, cell fate transitions, and disease development.

Key findings include the identification of glycine and proline residues as critical in phase separation, the impact of pathogenic mutations on phase separation, and the implication of these mutations in various diseases. The study provides a comprehensive analysis of the phase-separating proteome, emphasizing the role of key residues and their contribution to disease mechanisms.

Overall, the manuscript is well-structured and presents a significant advancement in understanding protein phase separation and its implications in health and disease. The methodology is robust, and the results are supported by extensive data and analyses, making this a valuable contribution to the field.

The authors of the manuscript have made substantial revisions based on previous feedback. They addressed key concerns such as improving methodological contrasts with existing methods, providing deeper structural insights, validating their PSPHunter model, and clarifying the connections between phase separation and disease. The responses seem comprehensive, with additional experiments and analyses, including figures and data to support their revisions.

Upon re-reviewing the manuscript, one area that could be clearer is the specific criteria and parameters used in the PSPHunter algorithm for predicting phase-separation key residues. While the manuscript explains the general methodology and the significance of the findings, a more detailed explanation of the algorithm's decision-making process could enhance the reader's understanding.

This includes information on the features considered by the algorithm, how it weighs different types of data, and the validation process for its predictions. Clarifying these aspects would strengthen the manuscript by providing deeper insights into the workings of PSPHunter and its application in different biological contexts.

As an example of useful clarification, in the section 'Construction of training and testing datasets' section, the method involves sampling the negative set 100 times and combining it with the positives, followed by a 70/30 train/test split. However, it is unclear whether the training is conducted on a single dataset and tested on 100 different datasets. Perhaps comparisons with available methods such as MaGS and PhasePred should be performed.

Reviewer #3:

Remarks to the Author:

The authors have addressed all my questions in detail. I recommend the manuscript to publish in Nature Communications.

One last comment on the web server of PSPHunter, maybe adding the sequence logo plot would help show the most significant mutations.

We appreciate the valuable feedback from the three reviewers. We've incorporated their suggestions and provided a comprehensive point-by-point response to each reviewer's comments. We also offer supplementary information and new experimental data to support our revisions.

Reviewer #1 (Remarks to the Author):

The authors have made great efforts in the revision and addressed the majority of my concerns. I am delighted to extend my support for the publication of their manuscript pending the addressing of the following minor points:

Response: Thank you for your positive feedback and support for our manuscript. We appreciate your thorough review and will promptly address the minor points you've mentioned. Your continued support is invaluable to us.

1. Authors should remove the term 'in vitro' in the legend of Fig.3i, as it is inconsistent with what is shown in the figure.

Response: We apologize for the oversight. We have promptly removed it to ensure clarity and accuracy in our presentation. Thank you for bringing this to our attention.

2. Fig.6e and f were not correctly referenced in the text 'Intriguingly, the Mut+IDR group exhibited a distinct restoration in the proportions of cells in S phase (Fig.6e.f).'

Response: Thank you for your valuable feedback. We have made the necessary amendments to the manuscript, including the referencing of Figures as (Fig. 6g,h) now accurately aligned with the correct figures in the manuscript.

Reviewer #2 (Remarks to the Author):

The manuscript "Precise prediction of phase-separation key residues by machine learning reveals the dysregulated landscape impacted by pathogenic mutations" presents a study in the field of biophysics and molecular biology. The authors developed PSPHunter, a machine learning algorithm, to predict key residues involved in protein phase separation. This tool is validated both in vitro and in vivo, demonstrating its potential in identifying disease-associated phase-separating proteins and understanding the mechanisms governing transcriptional control, cell fate transitions, and disease development.

Key findings include the identification of glycine and proline residues as critical in phase separation, the impact of pathogenic mutations on phase separation, and the implication of these mutations in various diseases. The study provides a comprehensive analysis of the phase-separating proteome, emphasizing the role of key residues and their contribution to disease mechanisms.

Overall, the manuscript is well-structured and presents a significant advancement in understanding protein phase separation and its implications in health and disease. The methodology is robust, and the results are supported by extensive data and analyses, making this a valuable contribution to the field.

The authors of the manuscript have made substantial revisions based on previous feedback. They

addressed key concerns such as improving methodological contrasts with existing methods, providing deeper structural insights, validating their PSPHunter model, and clarifying the connections between phase separation and disease. The responses seem comprehensive, with additional experiments and analyses, including figures and data to support their revisions.

Response: Thank you for your positive feedback on our manuscript. We're glad you found it well-structured and consider our work to be a significant advancement in understanding protein phase separation and its implications. Your recognition of our methodology and the extensive data supporting our results is appreciated.

1. Upon re-reviewing the manuscript, one area that could be clearer is the specific criteria and parameters used in the PSPHunter algorithm for predicting phase-separation key residues. While the manuscript explains the general methodology and the significance of the findings, a more detailed explanation of the algorithm's decision-making process could enhance the reader's understanding.

Response: Thank you for your valuable feedback on the clarity of the PSPHunter algorithm's criteria and parameters for predicting phase-separation key residues. We have revised our explanation in the manuscript accordingly.

First, we selected a truncated-unit value of 20 amino acids based on the average length of phase-separation proteins (~600 amino acids). This bin size is 1/30, which we consider relatively reasonable. It is worth noting that theoretically, a larger value for this parameter leads to a greater variation in the delta PSPHunter score. Additionally, experimental validation with a smaller number of truncated-unit of 3aa in the OCT4 protein has been reported (Wang et al., *Cell Stem Cell*, 2021). Users have the flexibility to adjust this parameter in our standalone version.

Second, to balance PSPHunter's sensitivity in identifying key residues, we aim to capture consecutive amino acids with the greatest impact on phase separation. To achieve this, we have set an empirical parameter, selecting 20-40 top-ranked key residues as candidates, aligning with 1-2 times the truncated unit. This balances user convenience and prioritizes crucial regions. We next connect consecutive amino acids in candidates to form key regions. Users can visualize these regions through our web-based service. Based on our comprehensive identification of key regions across the entire phase-separation proteome, we deem the overall quantity of 2-4 key regions to be reasonable.

To enhance understanding, we've incorporated a flowchart (Response Fig. 1) and provided expanded explanations in the manuscript. Additionally, all the aforementioned results have been updated in the revised manuscript (Lines 876-881 and 890-895). We believe these updates will greatly improve readers' comprehension of the PSPHunter algorithm's decision-making process. Thank you for highlighting this, and we appreciate the opportunity to improve our work.

Response Fig. 1 | Detailed strategy diagram for identifying key residues.

2. This includes information on the features considered by the algorithm, how it weighs different types of data, and the validation process for its predictions. Clarifying these aspects would strengthen the manuscript by providing deeper insights into the workings of PSPHunter and its application in different biological contexts. As an example of useful clarification, in the section 'Construction of training and testing datasets' section, the method involves sampling the negative set 100 times and combining it with the positives, followed by a 70/30 train/test split. However, it is unclear whether the training is conducted on a single dataset and tested on 100 different datasets. Perhaps comparisons with available methods such as MaGS and PhasePred should be performed.

Response: Thank you for your valuable feedback. Regarding how the algorithm weighs different types of data, we performed feature selection on all 123 sequence and functional features. These features were then ranked based on their importance value, and we assessed their contribution by progressively increasing the number of features. We observed that the model's performance reached its best when the number of features reached 60, with no significant improvements upon further increase. Consequently, we settled on 60 features for the final model. The detailed description of these criteria has been included in the revised manuscript (Descriptions in Revised MS, Lines 858-863).

Regarding the validation process for predictions, we apologize for any confusion caused. Our intention was to construct 100 training datasets and 100 testing datasets. We have clarified this in the revised manuscript to strengthen our statement (Descriptions in Revised MS, Lines 676).

Additionally, we greatly appreciate your suggestion to compare PSPHunter with MaGS and PhasePred methods. The performance results for PhasePred (AUC=0.877) and MaGS (AUC=0.886) both slightly lag behind PSPHunter (AUC=0.937) (Response Fig. 2). These updated results have been included in Fig.1b.

Thank you again for your valuable input, and we have incorporated these updates to enhance the clarity and robustness of our manuscript.

Response Fig. 2 | Adding MaGS and PhasePred to the comparison of PSPHunter on 100 independent test datasets.

Reviewer #2 (Remarks on code availability):

I checked parts of the code without running it locally.

Response: Thank you for your thorough review of the code, and we appreciate your effort in assessing it, even without running it locally. We have refactored the original code to ensure it is more adaptable for deployment on various machines. Should you have any specific concerns or questions about the code's functionality or availability, please do not hesitate to raise them, and we will provide assistance accordingly.

Reviewer #3 (Remarks to the Author):

The authors have addressed all my questions in detail. I recommend the manuscript to publish in Nature Communications.

Response: Thank you very much for your thorough review and positive recommendation for our manuscript to be published in Nature Communications.

1. One last comment on the web server of PSPHunter, maybe adding the sequence logo plot would help show the most significant mutations.

Response: We appreciate your suggestion regarding the PSPHunter web server. We agree that displaying the most significant mutations would enhance user-friendliness. In our updated version, we have added visual annotations for the top 3 regions. Additionally, we have implemented a zoom-in feature for users to explore specific amino acid mutation in more detail. Furthermore, users can download the raw data matrix to identify significant mutations based on specific values. Once again, we appreciate your suggestion.

Summary

Download the result

Note:

The lower the phase separation score (indicated by a bluer color), the greater the predicted impact of this mutation type on protein phase separation ability.

3. download raw data

2. zoom in and out

Response Fig. 3 | Enhanced features for displaying the most significant mutations.

Reviewer #3 (Remarks on code availability):

2. I have tried the webserver and it works fine. I can clone the Github code, but I was not able to run the code with the readme provided. It seems many of the paths in perl scripts only match the authors' workstation. The authors might want to check whether the Test case shown in readme works for a different workstation.

Response: We apologize for any inconvenience encountered while attempting to run the standalone version of our tool. In response to your feedback, we have refactored the original code to ensure compatibility with deployment on various machines. We appreciate you bringing this issue to our attention and thank you for your patience.

Reviewers' Comments:

Reviewer #2:

Remarks to the Author:

I am happy with the revised version of the manuscript.

REVIEWERS' COMMENTS

Reviewer #2 (Remarks to the Author):

I am happy with the revised version of the manuscript.

Response: Thank you very much for your positive feedback on the revised version of our manuscript. We greatly appreciate your time and effort in reviewing our work.